# Exploring the Impact of an Integrated Trauma-Informed HIV and Vocational Intervention for Black/African American Women Living with HIV

**DOI:** 10.3390/ijerph20176649

**Published:** 2023-08-25

**Authors:** Hsiao-Ying Chang, Vanessa Johnson, Liza Marie Conyers

**Affiliations:** 1Yang-Tan Institute on Employment and Disability, Cornell University, Ithaca, NY 14850, USA; 2Ribbon, Suite 200, 1300 Mercantile Lane, Largo, MD 20774, USA; vjohnson84bj@gmail.com; 3Educational Psychology, Counseling, and Special Education, Pennsylvania State University, University Park, PA 16802, USA; lmc11@psu.edu

**Keywords:** trauma informed, HIV and vocational intervention, Black/African American women

## Abstract

Given the increased recognition of the role of social determinants of health on the prevalence of HIV in the United States, interventions that incorporate and address social determinants of HIV are essential. In response to the health disparities facing Black/African American women living with HIV, HIV activists and mental health specialists developed an innovative integrated HIV prevention and vocational development intervention, Common Threads, that underscores and addresses key economic and other social determinants of health experienced by Black/African American women within a trauma-informed care (TIC) framework. This research study applied grounded theory methods to conduct a qualitative study of Common Threads based on interviews with 21 women who participated in the Common Threads intervention. Participants shared several critical aspects of program components that reflected the TIC principles, endorsing a safe environment, trust building, and a sense of belonging. These components also encouraged transparency and promoted autonomy. Additionally, participants shared perceived program outcomes, including changes of knowledge and skills in four considering work domains (i.e., medical, psychosocial financial/legal resources, and vocational) that facilitate health and vocational development.

## 1. Introduction

Many Black/African American women experience greater vulnerability to HIV which is related to the economic and social determinants of health associated with significant health disparities. Black/African American women are at least five times more likely to report HIV than women of any other race or ethnicity [1]. A review of HIV surveillance data from 2015 to 2019 indicates that Black/African American women accounted for more than half (42.1%) of new HIV incidences among women, while only 13.6% of the female population in the U.S. identifies as Black/African American [2,3]. The intricate interplay of racism, classism, gender power dynamics, and community conditions significantly exacerbate the vulnerability of Black/African American women living in low-income communities that are more vulnerable to HIV exposure [4]. According to Denning and Dinenno, HIV among heterosexual women who did not inject drugs and lived in geographic zones associated with poverty did not vary substantially by race [5]. However, the fact that a higher proportion of Black/African American women live in poverty (46%) compared to White women (10%) makes poverty one of the most significant social determinants of health for Black/African American women [6,7]. Research studies that examine the relationship between poverty and illness have provided sound evidence indicating that poverty is closely associated with the rates of disability and chronic health issues of all kinds, including unsuppressed viral loads [8,9]. Other health problems associated with poverty include but are not limited to a lack of access to health care, higher rates of cancer, a higher mortality rate, and an increased likelihood of other physical and psychological health issues [10,11,12]. Geography also intersects with poverty and the lack of access to health care. In 2018, 52% of all new HIV diagnoses in the United States occurred in the South in states with higher proportions of African Americans and an inequitable access to HIV prevention (e.g., health education, testing, and PrEP) and treatment services (e.g., transportation and insurance coverage) [13]. Medicaid coverage increases access to both general and HIV-specific prevention and treatment services. However, most states that have not expanded Medicaid coverage are in the South, contributing to disparities in healthcare access. The intersection of health and economic disparities leads to chronic financial stressors, which can also significantly impact HIV-related health management [14].

Research studies examining the roots of race-based HIV disparities among women suggest that the intersections of socio-structural impediments, including gender-based violence, mental health issues, substance abuse, and community and racialized trauma, also need to be considered to understand why Black/African American women are at greater vulnerability for acquiring HIV [15,16]. For example, Black/African American women disproportionately experience intimate partner violence and sexual violence [1]. This violence against women can hinder their ability to negotiate safe sex practices, seek HIV testing, or access healthcare support. Additionally, the threat of intimate partner violence may increase when women disclose their HIV status, potentially delaying their seeking of healthcare services [17]. The use of substances is also highly associated with trauma experienced among Black/African American women. Research studies have also shown that intimate partner violence is also closely associated with lifelong drug use, and affected individuals are at least twice as likely to engage in substance abuse and have an increased likelihood of unprotected sex [18]. Black/African American women living with HIV are facing a syndemic, a combined and mutually enhancing epidemic involving HIV, poverty, violence, substance abuse, and trauma. A previous study described a similar syndemic that includes food insecurity, intimate partner violence (IPV), poor mental health, and substance abuse, which negatively impact HIV risk behaviors and treatment adherence [19]. According to the syndemic theory, these interconnected challenges collectively contribute to the poor health outcomes observed among people with marginalized identities, including Black/African American women with HIV [20].

With increased awareness of the critical role of social determinants of health on HIV, national HIV strategic planning has become more focused on the need to integrate social and structural components into the HIV prevention intervention design [21,22]. Greater recognition of the multiple traumas that Black/African American women face due to the intersection of social and cultural factors with gender, race, and economic circumstances, and their impact on HIV vulnerability, has also prompted more attention to the development of trauma-informed care (TIC) interventions. TIC seeks to address the influence of trauma on both micro and macro levels and attempts to reduce the possibilities of re-traumatization. This approach is grounded in three key elements, which are “(a) realizing the prevalence of trauma; (b) recognizing how trauma affects all individuals involved with the program, organization, or system, including its own workforce; and (c) responding by putting this knowledge into practice” [23]. The TIC principles highlight the importance of ensuring safety, trustworthiness, transparency, peer support and mutual self-help, collaboration, mutuality, empowerment, voice, and choice, and addressing cultural, historical, and gender issues [23]. Growing evidence has suggested incorporating the TIC model into HIV interventions; nevertheless, HIV interventions that address trauma remain scant, and only a few interventions are rooted in TIC principles [24].

Over time, national HIV strategic planning has also increased the focus on the need to address the economic determinants of health and to increase access to employment services. Research indicates that vocational development and employment improve the quality of life and self-esteem for many people living with HIV as work can promote psychosocial well-being, enhanced financial stability, and opportunities for social interactions, increasing the quality of life and self [25,26]. Other motivations to pursue employment include gaining access to health insurance, social integration, and becoming role models for others [27]. Gilory and colleagues [28] found that full-time work is a protective factor of PTSD symptom development in women who have experienced intimate partner violence, highlighting the importance of stable employment in influencing one’s mental well-being. Research studies have also found that employment can promote a linkage to health care, medication adherence, and viral suppression for people living with HIV [26,29]. 

Although there is an increased recognition of the need for employment services for African American/Black women living with HIV, limited attention has been paid to the acute need for trauma-informed vocational services that are required. Research related to trauma-informed vocational services for any population is limited, but the recent increased focus on this topic among employment specialists and career counselors is promising. Given the ways in which trauma can impact vocational development, such as indecision, dissatisfaction, disruptions in learning, and the inability to envision success, it is critical to incorporate trauma-informed approaches to vocational services and for providers to be “trauma responsive” [30,31]. Nonetheless, Ibrahim and Kamsani [31] note that despite the understanding that integrating trauma-informed practices in career counseling can enhance clients’ personal and career well-being, mental health and career counseling are often siloed and addressed independently of each other, leading to major gaps in outcomes. Despite the potential benefits of employment, more than 41% of people living with HIV reported unemployment [32]. Some studies of employed PLHIV have found high rates of insecure work [33,34]. Rueda and colleagues found that a high level of job insecurity is significantly associated with depressive symptoms in people living with HIV [34]. Additionally, a scoping review examined employment and the HIV continuum of care and highlighted that unemployment correlates with a late HIV diagnosis, missing HIV medical visits, and delaying HIV care [26]. 

As a Black/African American woman living with HIV and a prominent advocate for improved HIV care and prevention for Black/African American women, Vanessa Johnson developed an intervention entitled Common Threads [35]. Although not initially conceived of as a vocational intervention, Common Threads was designed to help African American women to understand the multiple social, economic, and cultural factors that increased their vulnerability to HIV and to develop the skills to share their stories with other women living with or vulnerable to HIV to increase HIV awareness, engagement in care, and prevention. The Common Threads program was developed with full recognition of the need to unpack multiple layers of cultural, economic, intergenerational, and interpersonal trauma and the stigma associated with HIV that have increased the vulnerability to HIV among Black/African American women. The first phase included four days of training led by four Black/African American female trainers (i.e., a licensed professional counselor, a group facilitator, and two HIV peer specialists). The training reviewed common vulnerabilities to HIV, including the impact of individual and generational trauma through peer-led presentations and individual and group activities (e.g., trauma timeline and family tree) that helped participants explore the role of trauma on the acquisition of HIV for themselves and other participants. Through group facilitation and sharing, participants came to understand the “Common Threads” they shared as African American women that contributed to their economic and health status. 

As Common Threads was initially designed to be an HIV prevention intervention, a key component of the intervention was to facilitate the development of skills for participants to both present their personal stories to others (to reduce stigma and encourage engagement in HIV care and prevention) and to market their skills to be hired to present at community events and/or within the HIV service delivery system [27]. Although not initially conceived as a vocational intervention, the skill development aspect of Common Threads included key components of vocational training (i.e., public speaking, crafting effective messages, résumé writing, networking, and confidence building). As Common Threads’ participants began to develop and implement their skills, many requested additional training, which led to a second phase of Common Threads, which focused on developing micro enterprising skill development. This second phase entailed learning how to make jewelry and/or sew products and the skill to price, market, and sell these products on a local level. Eventually, a third “Marketplace” phase of Common Threads evolved where graduates of the Common Threads program sell their products at local and national conferences, which entails additional vocational skill development related to planning inventories, raising funds for travel, interacting with the public, and tracking inventories and income. Although Common Threads is now recognized as one of the only trauma-informed integrated HIV prevention and vocational development interventions, no research has been conducted to understand the impact of this intervention on participants or to increase our understanding of how the intervention has facilitated the vocational development of participants. 

To gain a deeper understanding of Common Threads as a trauma-informed integrated vocational development and HIV prevention intervention, the researchers conducted a grounded theory qualitative study to explore the experience of the participants with a focus on evaluating the impact of this intervention through the framework of the client-focused considering work (CFCW) model. This framework posits that individuals living with HIV consider a change in work status because of a change in major life events within one or more of four domains of influence: medical, psychosocial, financial/legal, and/or vocational. Within this framework, a change in any of these domains of influence can lead to the client engaging in the process of change associated with four stages in the transtheoretical model stages of the change: contemplation, preparation, action, and resolution. To evaluate the impact of Common Threads as a trauma-informed vocational intervention, the first research question explored program facilitating factors from a trauma-informed lens, and the second research question explored the impact of Common Threads on the four domains of influence noted in the CFCW. Here are the questions: How do women experience the Common Threads program’s components in relation to the trauma-informed care (TIC) framework?How do women perceive the impact of the Common Threads program on the four domains of influence of client-focused considering work (CFCW) (i.e., medical, psychosocial, financial/legal, and vocational development)?

## 2. Materials and Methods

### 2.1. Research Design

The current project uses a grounded theory method [36] to explore the experience of women participating in the Common Threads program and its potential impact on their health, mental health, financial stability, legal resources, and vocational development. The use of the grounded theory is appropriate to this study, given that the Common Threads program incorporates TIC and vocational rehabilitation components in which combined effects are not yet fully understood. Additionally, this study drew from the TIC model and the CFCW to aid protocol development and data analysis. TIC is a strength-based framework that emphasizes the pervasiveness of trauma and advocates for service delivery that utilizes the concept to identify, prevent, or intervene in traumatic events [37]. It recognizes that traumatic events are often caused by structural violence, such as racism, homophobia, transphobia, or sexism [38]. The TIC framework considers how a trauma-informed environment may impact participants’ participation in the program. The framework is useful as it considers the needs of Black/African American cisgender women living with HIV and the prevalence and impact of trauma. The CFCW acknowledges four domains, including medical, psychological, financial/legal, and vocational, that can serve as barriers and/or facilitators to vocational development and decision making.

### 2.2. Participants 

A total of 21 cisgender female participants participated in the current study. Participants were between the ages of 55 and 64, and 10 were diagnosed between 25 and 34. Twenty participants identified as Black/African American, and one identified as Latina. Thirteen participants pursued post-secondary education. More than half of the participants (*n* = 12) reported an annual household income of less than USD24,999. Seven reported incomes between USD25,000 and USD49,000, and two reported annual household incomes over USD50,000.

Seventeen out of the 21 participants were exposed to HIV through heterosexual contact, and one also discussed the possibility of injection drug use. Three participants were unsure how they were exposed to HIV; one shared that she contracted HIV because of being stabbed by a client. All participants completed phase one of Common Threads’ training, 17 continued joining the Microenterpise (M.E.) Circle, and 14 participated in the Market Place. The sample characteristics and sociodemographic status of the 21 participants are listed in Table 1. Twelve participants were working at the time of HIV diagnosis. About half of the participants (*n* = 11) were working during the interview. Most participants (*n* = 18) disclosed their HIV status to their friends, family, and others in their social network.

### 2.3. Data Collection

This study used the snowball sampling procedure. Snowball sampling refers to the recruitment method in which researchers obtain potential participants through referrals from other informants whose experiences are relevant to the study [39]. Potential participants were invited through the program coordinator and group leaders (i.e., recruiters). After potential participants showed interest in the program, recruiters connected participants with researchers to explain the study further and obtain consent for their participation. 

The current study used semi-structured interviewing techniques to learn about participants’ perceptions of Common Threads’ outcomes. Based on the responses to the initial interview protocol, additional questions and probes were incorporated to further explore the impact of gender and race on program participation. For example, the question, *how important was it for you to be part of a program consisting entirely of women?* was added to examine the impact of program structure. Each participant was asked to participate in a 60 to 90 min in-person or phone interview. Verbal informed consent was received prior to the interview. The interview began with a researcher’s introduction and the greeting of the participant. The list of questions explores the participants’ experiences in Common Threads regarding program satisfaction, the impact of the program environment, and activities on physical and medical, psychosocial, financial, and legal, as well as vocational development. 

### 2.4. Data Analysis

This study used word-processing software and professional services for interview transcriptions. Each interview transcript was imported into and analyzed through an online software, Dedoose, Version 8.0.35 [40], to gain insight into interview content through coding, memo writing, comparing and connecting, and developing larger themes [41]. The interview content was analyzed utilizing grounded theory analysis (GTA) procedures, as suggested by Glaser and Strauss [42]) and Saldaña [43]). This involved iterative coding cycles, beginning with a low-inference, general coding process (open coding) and subsequently refining through theoretical, selective, pattern, and axial coding.

Three researchers were engaged in data analysis using GTA procedures to code interview contents, identify emerging data patterns, generate themes, and conceptualize relationships among themes [44]. As data collection proceeded, the process of data analysis was ongoing and continuously refined through theoretical, selective, pattern, and axial coding in a second phase. We started with a few transcripts. We compared initial codes, which were critically examined, re-coded, and reviewed as the data collection continued. Discussions with the research team, faculty, and peers were integrated into the process, fostering the refinement and redefinition of codes. Codes with similar discrete ideas were then grouped into a category with additional discussions. An example of a category is Common Threading, which includes codes like sharing skills, networking, connecting, sharing resources, and sharing stories. The researchers engaged in ongoing discussions on codes and themes until new information and concepts could not be found. 

### 2.5. Ensure Trustworthiness

Measures were taken to maintain the rigor of the qualitative research and enhance the trustworthiness of this study. First, this study included a pilot study to ensure the feasibility of the research ideas and methods. This also informed and strengthened the research design and modifications. The pilot study was useful to help the researchers narrow down the focus and establish a solid understanding of how to operationalize the concepts of TIC and considering work domains [45]. Secondly, the research team members verified the accuracy of the transcriptions after the recordings were transcribed into Word files. Moreover, this study used investigator triangulation to understand the data comprehensively. Investigator triangulation emphasizes that the understanding of the data is confirmed by different observations and perspectives of two or more researchers [44]. The research team members were asked to code transcripts after the primary research developed the code structure. The purpose of involving multiple researchers is to confirm the structure and support the trustworthiness of the data analysis and interpretation. All three researchers coded the transcripts independently and compared the theme codes until a consensus was reached.

## 3. Results

### 3.1. Trauma-Informed Program Facilitating Factors

The first part of the results addresses research question one: How do women experience the Common Threads program’s components in relation to the trauma-informed care (TIC) framework? The participants shared several critical aspects of Common Threads that reflected key components of TIC: safety, trust and transparency, choice and empowerment, and choice and empowerment, including (1) shared commonalities (i.e., as women living with HIV and a history of trauma) with program participants, (2) gender-specific and peer-led programming, (3) building family trees and personal timelines, (4) practicing storytelling skills, (5) gaining knowledge on HIV, HIV-related resources, and health benefits, and (6) vocational rehabilitation options. The participants’ comments highlighted how these fundamental aspects facilitated a safe environment, trust building, and a sense of belonging, empowerment, encouraged transparency, and promoted autonomy, which aligns with the TIC principles. Table 2 includes example quotes of each theme related to trauma-informed program-facilitating factors.

#### 3.1.1. Safety

Many participants felt protected against psychological and physical harm at Common Threads’ training, which helped them participate in training activities, build relationships with other women, develop a communal identity, and further benefit from the Common Threads program. Participants noted a couple of factors were fundamental in building a safe space, including shared commonalities (i.e., as women living with HIV and a history of trauma) and gender-specific and peer-led programming. 

The sense of safety was facilitated by participants’ recognition that others in the group would understand what they had gone through since participants shared commonalities. As one participant shared, “It was a good thing to meet other women to know what they are going through, and [they know] what I am going through (P3)”. Another participant commented,

[Women from Common Threads] didn’t sugarcoat nothing for me. But when I hurt, they held my hand. They never told me, ‘Don’t cry’, [or] ‘Stop that crying’. That’s what my boyfriends told me when I cried when they hit me. [Women from Common Threads] walked me through every difficult step of the way, every step.(P21)

The participants felt more comfortable sharing when they were with their peers. As one participant shared, “[The peer leader] knew more issues about [HIV], and they and I were on the same stuff (P5)”. The gender-specific component also helped to build a safe environment. Most participants noted that they were more likely to share because the program was a gender-specific and peer-led intervention. The comments on gender-specific structure include that the participants felt they were not constrained by stereotypical gender roles and appreciated not having to interact with men, a group with which they associated many traumatic experiences. As one participant shared, 

Common Threads understands the importance of women being with women and hearing women connect that way. Because I have other experiences with groups [that] had male consultants in the room, and I always felt the dynamic didn’t go as far as it needed to go because of that presence.(P3)

#### 3.1.2. Trust and Transparency

Feeling safe perhaps instilled trust in other members. The participants believed they could rely on their peers to empathize with their stories, keep them confidential, or provide support when they reached out. Some participants noted that the acknowledgment of mutual trust was important in relationship building. For example, one participant explained how she rebuilt her trust in people, stating,

I have a problem with trust, but I could trust them because they were saying things they felt like they could trust me, so I had to give it back to them. We learned to trust one another, and we all are dealing with the same thing.(P6)

Storytelling activities facilitated the group process. For example, participants commented that they felt the facilitators and their peers were honest in sharing their stories, which initiated this sense of belonging, as one participant shared, remarking, 

The clinician and [group leader] led by example and got personal. They did an in-depth presentation on their past with their own family from their family tree. They were very transparent with us, which was so rare, and let us know about the abuse and molestations, the diseases, and how they trickle down to them. By letting me see that and their emotions, it just was like, ‘Wow’, it was a lot of points that I could identify.(P9)

Many participants noted the stigma associated with HIV and discussed the feelings of loneliness and isolation they experienced when they were first diagnosed with HIV. One participant shared her experience when she was first diagnosed with HIV, saying, “It had me think of, I could not be around other people”. This participant felt that Common Threads provided her with a sense of belonging and further facilitated a better sense of self-worth. 

#### 3.1.3. Choice and Empowerment

By hearing different personal histories and engaging in discussions, participants felt informed of different choices and were better at gauging situations and making decisions. As one participant noted, “[Common Threads] allows me freedom of choice to make the right decision, to do the right thing for the right reason”. The participants also shared that because of the interactions with other participants, they had more knowledge of HIV and benefits, laws, and information on resources, which expanded their awareness of the possibilities available to them. Vocational training, experienced through activities such as selling products that increase one’s financial resources, also plays a role in facilitating the development of psychological and economic autonomy. One participant commented, “[Common Threads] educates the women and empowers them to learn how to be independent and self-sustaining. We need to have a class on teaching women how to extend their money. How to budget (P12)”.

#### 3.1.4. Culture and Race

During the interview, the participants were asked if it was helpful to be in training with women of the same race. Some participants emphasized the importance of training with all Black/African American women. One participant acknowledged that some Black/African American women might feel less comfortable sharing in a mixed-race group. Another participant discussed the role of sharing culture-specific language, and a shared understanding of communication expression is essential. The participants also discussed their experiences and the impact of racism. For example, one participant shared a story that revealed how racism had affected her,

[When I was ten years old] I didn’t think I was different until one day, a white lady walked up to me and said, “n-----, what are you doing here?” That was the first time I saw white women, and I realized that I was different. My frame of reference was that I’m in the wrong place; I should never be in a place where white folks are.(P21)

This participant explained the sense of lower self-esteem and not fitting in the White world (i.e., not being accepted) because they often seem affluent. Although she later discussed that meeting other White/Caucasian participants in other HIV workshops and Common Threads has helped her develop a new frame of reference towards being Black/African American and the relationship with White women. Her comment suggested the importance of addressing race in HIV intervention for Black/African American women and the benefit of having race-specific groups. While most participants participated in Common Threads with other Black/African Americans, many discussed that they are now open to having a mixed-race group and noted the benefits of learning from women with HIV of differences in race, ethnicity, and even nationality.

### 3.2. Perceived Outcome Related to the Client-Focused Considering Work Domains

The results related to the second research question—How do women perceive the impact of the Common Threads program on the four domains of influence of the client-focused considering work (CFCW) model (i.e., medical, psychosocial, financial/legal, and vocational development)?—are reported below. This section defines each outcome area, followed by the main themes and subthemes of the outcome area. Table 3 outlines the themes and subthemes of the perceived program outcomes. Table 4 includes example quotes related to the themes of the perceived program outcomes on the considering work domains.

#### 3.2.1. Medical Domain

The participants discussed how Common Threads had influenced their health, and two themes emerged from the interviews: Manage Health and Improved knowledge and resources. Specifically, the Manage Health theme refers to the participants’ increased awareness of their health status, their ability to care for themselves, and their consideration of medication adherence. Improved knowledge and resources refers to the participant attaining the knowledge and resources necessary to promote their health. 

##### Manage Health

Many participants discussed their substance use, abuse, and nonexistent self-care histories. Additionally, all the participants reported having an undetectable viral load status at the time of their interviews. Therefore, the interviews focused more on how the training had influenced their health management, if it had at all. Many participants noted that the training had reinforced and helped them continue practicing behaviors promoting and maintaining health. For instance, as one participant shared, “[Common Threads] lets me know that I have to keep participating in my health and well-being by keeping my appointments and being honest with my doctor. It lets me know that I am loved (P9)”. Another participant noted the reality of medication adherence, saying, 

We all go back and forth with this. Most people would say, ‘Oh, I don’t want to take this medicine I don’t need’, but there are certainly necessary times when people do need it. [Common Threads gave me] the motivation that if I keep building my body and keeping it healthy, I can continue to do these good things with other people who are producing. (P3)

Another participant discussed how she regained control by managing her health and taking medication as she shared, “[I learned] you have to stay on top of it. You have to take your medication and go to the doctor. You can’t let it beat you. You have to take control of it (P1)”. These quotations suggest that the participants felt responsible for maintaining their good health as a result of the training. It is also apparent that the idea of health management was not limited to managing HIV-related health concerns but also included other physical health concerns, such as diabetes. 

##### Improved Knowledge and Resources

Some participants discussed the training content in detail, including the parts during which they had learned about HIV and gained resources. One participant shared her experience participating in training programs other than Common Threads, commenting,

My experience with other HIV support groups was just getting together, eating food, and talking, but they were not talking about anything [I needed]. I wanted to talk about medical directives, the difference between them, and how to get help finding an insurance company that would take me with my preexisting condition. I wanted to learn more about my disease, the medication, and how I could help my body and my immune system to be top-notch and things of that nature. (P9)

Another participant commented that she had gained confidence and discovered strategies for acquiring information as she stated, “I can ask questions without feeling like I’m dumb. I write the questions that I want to ask before I go to an appointment. It’s helped me prepare myself for what needs to be done (P5)”. Some participants shared that they could share the knowledge that they had gained from the training with their peers or family members. For example, one participant noted, 

When I meet young women with children, I would introduce them to protection, like a condom, or get them to have a pap smear every three months, [and] let them know] there are certain things you can do to check up on yourself.(P6)

#### 3.2.2. Psychosocial Domain

Psychosocial outcomes refer to a perceived change in personal development that promotes social and psychological well-being because of Common Threads. The participants discussed how Common Threads had influenced their psychosocial development, and three themes emerged from the interviews: Building a community, Improving mental health, and Improving psychosocial skills. There are several subthemes of the three major themes that are discussed below.

##### Building a Community

The theme of Building a community refers to participants’ engagement in creating an environment with a shared understanding of the issues they face and a common goal, which is to support women living with HIV. Through the peer group intervention, participants were able to build connections with other women living with HIV. Many participants felt they had to deal with HIV alone when they first received their diagnoses. They struggled with stigma, mental health, and medical issues and felt closeted. One of the keys to community building is establishing connections. Many participants shared that they realized they are not alone beyond by simply being with a group of women living with HIV but by also hearing other women share their stories and seeing themselves in other people’s stories. As this participant remarked, 

[Common Threads] has allowed me to realize that we may look different, but we have more in common than not. We may have reached this place through different roads, arenas, behaviors, or incidents, but we’re here now, and the common thread is that you don’t have to go alone.(P9)

While the stigma associated with HIV can isolate individuals, the group experience helped the participants reconnect with others. A new group identity, Womanhood, was identified, capturing the sense of belonging and illustrating the value of the community. The participants used different terms to describe Womanhood, including comrades, sisters, sisterhood, and Common Threaders. They ascribed new meanings to these terms, their relationship with each other, and what it means to be a woman living with HIV. As one participant shared, “We are women, and, we are positive. [This] brings us together and makes us feel like we’re useful (P16)”.

Participants discussed wanting to pay it forward or give back because they realized how they have benefited from being a part of the community. They discussed taking different actions, including helping others, becoming a peer specialist, joining the community board, speaking at conferences, and participating in other support groups so they could share what they had learned from Common Threads. As one participant shared, 

I’m on the board of directors and would go out in the community. I was talking at the [HIV] summit for the first time, I was nervous, but I told my story. I understood a lot of women were going through mental and physical abuse with their loved ones. [Common Threads] made me more aware of reaching out instead of closing in. It gave me a sense of joy and peace to say, ‘I can do this. I can teach someone else’.(P13)

Participants highlighted that disclosure is another strategy for building a community. The participants shared that because of the trauma timeline activity and storytelling skills that they learned, and by hearing other people share their stories, they felt encouraged to share their own stories. Here is an example quote describing how Common Threads reinforced disclosure:

[Common Threads] allowed me not to be afraid. I hope the information [I share] can save someone else’s life. [Before the training] I would have never told someone to go to the clinic and speak with someone about medicine or about getting the medication that you need to help you live a life with this diagnosis. Now, I won’t think twice about it. If somebody wants to look at me differently, then that’s fine. It doesn’t change who I am, and it doesn’t change what I’m on this earth to do. (P9)

There is sustained support from the community in that participants shared that they were still involved with other women from the Common Threads’ training. When the interviewer asked the participants about their relationship with other women in Common Threads, the frequency of engagement ranged from daily to monthly or as needed. Some participants elaborated on how the sustained support affected their medical and mental health and consideration to work. For example, one participant described, “When I’m going through a bad time, I can pick up the phone and call somebody or e-mail them, and I’d continue to feel empowered (P17)”.

##### Improving Mental Health

Participants discussed that their mental health had improved after participating in the program. Two subthemes, Building Confidence and Having Hope, were identified by an amalgam of factors that had led to improving participants’ mental health. The participants discussed different areas of personal development, such as improving communication and storytelling skills, understanding the impact of trauma, and developing strategies to support oneself, which have helped build confidence. For example, the participants discussed that the vocational intervention (i.e., the M.E. Circle) provided training on building vocational skills and engaging in meaningful interactions with others, further promoting self-confidence. One participant stated, 

We made jewelry and sold the jewelry. We were telling the people why we were selling it and what the benefits would go to. I learned when someone asked me, ‘Well, are you positive?’ [I am] more confident to be okay with who I am and to talk to other people about it if they should ask me any questions.(P10)

Participants also talked about how their increased knowledge of HIV helped them reshape their self-worth. One participant commented, “[I learned] we aren’t our diagnosis, that we’re just regular people with a blood-borne disease that can be monitored, and we can live on as normal people”. Understanding and accepting that an HIV-positive status “is not a death sentence” and that a diagnosis does not define a person was crucial for the participants. Hearing their peers share their experiences living with HIV helped the participants to feel empowered. The participants expressed having hope and a can-do attitude after they discussed the influence of hearing other women’s stories and seeing other women’s timelines. As one participant shared, “[We have] seen these women survived and thrived. And we can do that also (P7)”. Including peer leaders also helped people to see the path (i.e., how to) to improve their health practices and quality of life. For example, as one participant shared, “By them doing what they did, I was able to mimic [what they did] (P4)”. 

##### Improving Psychosocial Skills

The participants discussed different areas of improvement in psychosocial skills, including managing stress and enhancing communication skills. Participants shared that they had learned some stress management skills, such as breathing techniques. For example, as one participant shared, “[Common Threads] taught us how to just breathe and how to do exercises to relieve the stress (P11)”. This participant shared that these exercises had allowed her to “step back and look” at a challenging situation from a different perspective, reducing her emotional distress and intensity.

Many training activities also facilitated the development and improvement of communication skills, such as reading, creating the trauma timeline and family tree, storytelling, and participating in the M.E. Circle and marketplace. Participants shared that they learned how to express their feelings and deliver and receive information effectively through verbal communication in different interpersonal situations, especially when broaching difficult topics. For example, as one participant shared, “I was already out speaking, and [Common Threads] gave me more insight and education on how I’m supposed to go about speaking (P2)”. As another participant shared, 

[Common Threads] allowed me to have resources, to know where to get help, how to ask for help, the correct place to go for help if needed, and to advocate for myself when I have doctor’s visits, to advocate for my rights, and not to be a victim.(P9)

Another participant noted the importance of gaining assertive communication skills during the training, as she stated, 

When I grew up, girls never said no. Now when I think I am being abused, I learned the power of saying, ‘No, I am not doing it. It doesn’t feel good, no’. I learned that saying ‘no’ is okay.(P15)

#### 3.2.3. Financial/Legal Domain

Financial/Legal outcomes refer to a perceived change in personal development that promotes financial and legal well-being because of Common Threads. An example of legal well-being may relate to receiving health care benefits. A theme, Increasing financial/legal resources, was identified during the participants’ interviews, as many participants discussed how they could sell products they had made as an additional source of income and increased resources to understand social security benefits. For example, a participant noted, 

[Common Threads] brought about ways to produce income because a lot of us may not be able to work in the secular world, but we still need to support ourselves. I was so impressed [to learn how] that a person could be financially stable. Take the skills that you have and transfer them and become your own boss.(P9)

Participants shared that they have been selling their products and increasing their income. One participant commented, “In the wintertime, I crochet [and sell], I have a little $25 here, $25 there, and that helps with the food (P18)”. Participants also discussed the resources they had gained to support their understanding of health benefits. For example, as one participant shared, “Common Threads gave us a flyer on health and information on public benefit, and we talked about it because I wasn’t the only one there dealing with it (P11)”. Another participant commented that she learned how to navigate the social security benefits system as she continued working. 

#### 3.2.4. Vocational Domain

Vocational outcomes include both work- and education-related findings. When asked about how Common Threads had impacted the participants vocationally overall, many participants discussed the knowledge and skills they had gained that directly and indirectly enhanced their vocational outcomes. These types of knowledge and skills include, but are not limited to, understanding the impact of trauma, HIV-related knowledge, health management, self-care, storytelling, and psychosocial skills. These skills have facilitated the expansion or strengthening of participants’ vocational skills or have led them to consider work or education. One participant shared how Common Threads helped her to improve her reading skills, stating, “I didn’t finish school, so I have problems with reading, and it gave me more confidence that I could actually read (P6)”. Another participant shared that she used Common Threads’ tools to train others, stating, “I still use it when I’m training others. I’ll do the timeline, and I still use the yarn that we had to throw to another sister with a word of encouragement (P1)”.

The discussion of newly acquired vocational skills also focused on skills that the participants had learned from the M.E. Circle, such as crafting, crocheting, jewelry making, and résumé writing, as well as communication and networking skills. One participant commented on the impact of this vocational training, saying, 

[M.E. Circle] helped us to prepare for seeking employment services and taught us that we could be empowered by other women and be self-sufficient. Common Threads teaches you to love yourself, and then you can start manifesting the other stuff in your life. You can come out of fear, and you can come out of shame and guilt.(P16)

Many participants also noted they had experienced expanded vocational opportunities because of Common Threads’ training. One participant noted that many of her peers had obtained jobs, such as becoming beauty-product salespersons, after participating in Common Threads. As another participant expressed, 

[Common Threads] opened our eyes, opportunities, and doors for us to find out what it would feel like to make a product, advertise and actually sell that product. I do it now on my own with making jewelry and crochet. (P9)

Participants also discussed getting a job as a peer support specialist or entrepreneur or returning to school. One participant explained that being an entrepreneur meant “being self-sufficient that you have a skill or a trade that someone will want to purchase”. 

Many participants expressed the desire to participate in vocational activities. For example, one participant noted, “[After Common Threads, I want] to go back to either catering or selling certain things, doing flea markets or having the marketplace here or whatever to pay for my education (P12)”. As a result of Common Threads, the participants also experienced greater clarity in recognizing the paths they needed to take to achieve their goals. As one participant commented,

I wanna go back to work. Now, I want to work for myself. I want to be an entrepreneur and teach what I do. [I need] funding and know how to write a grant. I have written a business plan but need a mentor to show me how to go about it. Someone to guide me to the right path that has done something like that before.(P18)

It is also clear that participants had more resources to evaluate their health and psychosocial well-being in light of their vocational decisions. For example, one participant expressed that she had considered vocational activities about her health status, as she noted, “[I can] maybe get a better job and work a few more hours if my energy allows me to (P10)”.

## 4. Discussion

Black/African American women living with HIV experience a multitude of traumatic life experiences that greatly increase their vulnerability to HIV. Although poverty has been identified as a key social determinant of health for this population, few, if any, HIV prevention interventions have been evaluated to explore the impact of applying a trauma-informed approach on participants’ vocational development. The purpose of this study was to use grounded theory’s qualitative methods to gain a deeper understanding of Common Threads, a trauma-informed integrated HIV-prevention and vocational-development intervention. In response to the two main research questions, this study makes several contributions to research related to the social determinants of HIV among Black/African American women. First, this study’s findings describe the ways in which the Common Threads intervention applied a trauma-informed approach. Participants reported that several Common Threads’ components (e.g., peer-led, gender-specific, group timeline, and family tree) facilitated a sense of safety, trust, autonomy, and empowerment associated with trauma-informed care (TIC). Second, this study also expands our understanding of how participants perceived the impact of Common Threads on the four domains of influence critical to vocational decision making through the lens of the client-focused considering work (CFCW) model. Overall, the findings support the importance of Common Threads as a unique and innovative intervention and the need for additional research in this area. This discussion will elaborate on the findings associated with each of the key research questions in greater depth and discusses implications for research and practice.

### 4.1. The Common Threads’ Trauma-Informed Care (TIC) Program Components

Consistent with the research literature documenting the high prevalence of trauma among Black/African American women living with HIV [46,47], Common Threads’ participants revealed significant histories of interpersonal and intergenerational trauma. The research literature underscores that a diagnosis of HIV is traumatic in and of itself and often compounds and magnifies trauma [48]. As noted in our findings, many participants described shock, distrust, guilt, and isolation from their friends, families, and communities after they received an HIV diagnosis and feared disclosing their HIV status. The research literature notes that a history of adverse experiences or the experience of an HIV-positive diagnosis can make a person feel disempowered and disconnected [48,49]. Trauma-informed care suggests that rebuilding a sense of safety is essential to restore trust, which is the first step of recovery before working on goals [50].

As we consider the context for supporting Black/African American women with HIV, it is important to widen the lens beyond the interpersonal and family context. Trust and safety building need to be considered within the context of racial and gender-based trauma on a social and societal level [51]. When exploring the impact of participating in an intervention specifically designed for Black/African American women, one participant recounted the profound shock from being targeted with a racial slur as a child, which led to her early understanding that she was unwelcome in spaces occupied by White individuals. This narrative underlines the urgent need for interventions that address racial trauma. With respect to gender, participants also expressed the benefits of an all-female intervention, with one participant noting less satisfaction with the dynamics in other interventions that included men. Research provides support for the benefits of race and gender-based interventions for African American women living with HIV [52,53]. 

Throughout American history to the present, Black/African Americans have endured significant levels of harassment and discrimination across a variety of domains (e.g., housing, unemployment, and law enforcement) [54]. Extensive research studies have documented the historical mistreatment of Black/African Americans withing healthcare settings that have fueled distrust in health care for many [55]. Common Threads’ participants conveyed the importance of having the opportunity to discuss racial trauma and mistrust of the medical system among peers with similar experiences. This finding underscores the importance of having access to culturally sensitive interventions such as Common Threads, where the impact of individual and collective racial trauma can be acknowledged and validated. Comments from participants highlighted that the impact of systematic racism is evident in everyday life for many Black/African Americans. Creating a safe environment for Black/African American women living with HIV is one of the first steps to facilitate engagement and recovery [56]. These findings indicate that an intervention structure that acknowledges racial trauma is a critical component of a trauma-informed intervention for Black/African American women [23]. 

Participants also highlighted the importance of having a woman-centered approach to training and discussed different roles women hold; women’s unique needs in the financial, vocational, health, and mental health domains; and power dynamics between individuals of different genders. The importance of having a woman-specific HIV intervention has been consistently addressed in research [57,58]. A woman-specific component relies on more than simply the inclusion of exclusively female stakeholders. A systematic review of woman-specific HIV services identified four key aspects that successfully engage women living with HIV: (1) establishing a safe space with respect and acceptance; (2) promoting social interactions among peers; (3) emphasizing women’s autonomy; and (4) promoting self-determination [59]. Participants in a different qualitative study also highlighted the need for more woman-specific services to address HIV stigma and HIV-related healthcare needs [60]. A key component of Common Threads was the modeling of the open disclosure of past traumatic events by Black/African American woman peers with HIV. Both the peer-support and woman-specific components of the training supported trauma-informed values [61]. Peer-based HIV prevention interventions strengthen psychosocial support, engagement, and long-term retention in HIV care [62,63,64], which aligns with the findings of the current study.

Storytelling activities were also a critical facilitator to the recovery process. Through storytelling activities, the participants pieced together their trauma stories and learned how to better understand emotions, organize their thoughts, and communicate with others. The participants expressed that they had been able to reconstruct their knowledge and views of themselves, their personal experiences, and HIV, which led to their developing health management strategies and vocational goals and establishing a supportive community. The stories that people tell convey their worldviews and self-image [65]. For example, many participants felt that they had to manage HIV on their own, and they believed that other people could not be trusted. Such stories may evolve with time as individuals interact with different people and environments and gain more information and resources. Therefore, in the process of storying and re-storying, people can use their increased knowledge and resources to ascribe new meanings to experiences [65,66].

Studies examining disclosure have consistently found that an increased willingness to disclose is closely related to perceived social support [67]. Storytelling has been used extensively in health and psychosocial interventions and is important in promoting health, self-esteem, interpersonal relationships, and social support [68,69]. It is important to note that having a support system is essential for storytelling because revealing personal histories can make people feel vulnerable and elicit intense and distressing emotions and responses, especially when shared memories are traumatic [50].

### 4.2. Implications of Common Threads on Considering Work Domains

Traumatic experiences are known to have a negative impact on vocational development [70]. Failing to acknowledge the impact of trauma and to incorporate TIC principles in service delivery may impede the primary goal of vocational rehabilitation services, which is to facilitate employment [71]. Incorporating TIC principles into vocational interventions may be particularly relevant for people of color with disabilities who face additional barriers in accessing vocational rehabilitation services. Ji and colleagues [72] found that Black/African Americans were less likely to be screened for services, receive job training, and obtain gainful employment. Another study found that “refused to cooperate” and “unable to locate” were the main reasons for case closure among this population, which is likely to be impacted by financial and housing instability [73]. Given the high rates of trauma reported by Common Threads’ participants and overall limited research on trauma-informed vocational counseling, these negative vocational rehabilitation outcomes are not surprising. These studies warrant the need to expand the understanding of trauma-informed vocational interventions, as unstable housing and income can increase exposure to traumatic experiences [31,74]. 

The current study used grounded theory’s methods to better understand the impact of a TIC intervention on vocational development through the lens of the considering work model. The identified program outcomes provide early evidence supporting vocational development and reduce barriers that the Common Threads’ participants discussed through changes in one or more of the key domains of influence described in the client-focused considering work model. For example, the participants emphasized community building as well as improvements in health and mental health management and psychosocial skills. The HIV-focused literature has consistently highlighted the negative impact of HIV stigma on social and vocational participation [75]. This stigma often manifests as discrimination against and the isolation of people living with HIV. The participants shared similar experiences of significant isolation and a lack of support from their families, churches, and the broader Black/African American community. A common refrain among the participants was the desire to belong or to be with other women “just like me”. 

Community building is more than simply gathering similar people together, however. Weil [76] defined community building as the “activities, practices, and policies that support and foster positive connections among individuals, groups, organizations, neighborhoods, and geographic and functional communities” (p. 482). Community-\ building has also been conceptualized as the process by which people take control of their lives and environments [77]. Based on the various definitions of community building and feedback from participants, this study describes a community as a place where individuals feel a sense of belonging, trust, and a shared understanding of needs that they work together to fulfill.

Constructing a shared identity was essential to forming a community. Identity becomes salient when a person is engaged in more relationships within his or her environment and experiences increased commitment to the group [78]. Identity development is associated with changes in self-esteem and self-efficacy, which have consistently been found to impact people’s mental health, behavior, and decision making [79]. Previous research studies have found that people more engaged in HIV identity development are better at managing their health and maintaining relationships and tend to have more robust social support networks [79,80]. Career counselors taking an integrated trauma-informed vocational rehabilitation approach may support clients with trauma in building or identifying a supportive community [31].

The enduring influence of Common Threads is reflected in participants’ remarks regarding the ongoing social connections and support even after the program has ended. Participants reported continuous engagement and supportive relationships with other women from Common Threads over time due to lasting relationships established among some participants. The sustained support continues to influence participants’ health management, mental health, and engagement with the HIV community [81]. A study that examined the connection between individual social capital and the HIV continuum of care found that people living with HIV who benefited from greater social support and wider social networks were more likely to adhere to their treatment and achieve viral suppression [82]. The current study shows the importance of HIV interventions to support stable and long-term relationship building among the participants and communities. Additionally, Common Threads acts as a temporary social network. Within this social network, participants are given the opportunity to reconnect with an existing social network (e.g., family, friends, and associates), or they create a social network post-Common Threads with participants within the group. An extended social network appears to be important in facilitating continuous improvement in psychosocial functioning and health management, as well as the expansion of vocational interests and consideration of work.

The current study highlighted that it is crucial to have a comprehensive understanding of available resources in order to make informed decisions. This includes knowledge and resource access enhancement in medical, financial/legal, and vocational areas. For example, knowledge of health-related information is significantly associated with health outcomes and access to health care services [83]. Participants noted that they had gained HIV-related knowledge and resources and highlighted that the program had reinforced their commitment to self-care practices and medication adherence. They also commented on their increased information-seeking skills, noting that they “can ask questions” if necessary. 

Access to information can be particularly challenging when public systems, such as healthcare, have historically failed to adequately communicate information and consider the needs of individuals in Black/African American communities [84]. Black/African Americans have reported less access to healthcare and low utilization of health services [85]. In fact, the lack of access to healthcare services, exacerbated by the COVID pandemic, has led to heightened disparities in health outcomes and increased mortality rates among people from historically marginalized communities [86]. Moreover, with historical and ongoing challenges, a syndemic environment has made Black Americans particularly susceptible to the severe effects of COVID-19 [87]. Services’ lack of cultural competence remains an issue. Prior literature has suggested racial disparities in the quality of patient–provider interactions, such that the characteristics of the patient influence the level of information, type of referral, and quality of healthcare received [88,89]. Therefore, increased knowledge and skills for communicating with healthcare and service providers are important for people living with HIV, as many want to be involved in the treatment decision-making process [90]. By being trauma-informed and culturally sensitive, Common Threads prepared participants with knowledge and skills in four considering work domains that further facilitated the engagement of vocational activities, such as résumé writing or the M.E. Circle.

Vocational training was emphasized in Common Threads. For example, Common Threads includes microenterprise training in response to the prevalence of poverty among Black/African American women living with HIV. Our sample’s demographics were living below the poverty line of a three-to-four-person household, as most of our participants had an approximate household income of less than USD24,000. Although the goal of Common Threads was not to increase participants’ incomes and employment rates, many participants discussed increased income sources and vocational development opportunities because of the microenterprise training and marketplace. A participant commented on her increased knowledge of navigating the social security system after she had developed her own business. These comments support the importance of addressing the financial and legal barriers to considering work, as women living with HIV have reported concerns regarding the loss of health benefits caused by the decision to work [27]. 

Additionally, because of their HIV-positive status, women may lose social networks that could support their vocational development. Despite these challenges, participants shared their improved vocational skills and discussed their increased awareness of different vocational possibilities. The improved job skills and sense of self-efficacy related to job exploration further supported the vocational development of participants. These findings regarding vocational development are consistent with those of other HIV interventions that integrate vocational skills training and health education, specifically improved self-efficacy and belief in one’s ability to work [91]. The current study helps to better understand how the considering work model reflects a trauma-informed approach to vocational intervention for Black/African American women with HIV.

### 4.3. Implications for Practice and Future Research

This study suggests several avenues for service providers and future research. First, this study highlighted the importance of trauma-informed program structures and components that can establish trust, safety, transparency, autonomy, and acceptance, which are essential to promote positive program outcomes. In light of the growing recognition of the widespread nature of trauma and its potential to increase the vulnerability of Black/African American women to HIV acquisition, service providers could benefit from integrating trauma and trauma-informed care (TIC) training into their standard curriculum. Shaewitz and Yin [92] found racial disparities in state vocational rehabilitation systems, with Black/African Americans having the highest rate of applying for services but being least likely to be deemed eligible compared to White/Caucasian applicants. This finding suggests that not only is there a need for more trauma-informed and culturally sensitive interventions, but additional training is also required for vocational rehabilitation and career counselors to become more competent in delivering culturally sensitive services in a trauma-informed manner [31,93]. It is important for service providers to thoroughly understand the effects and manifestations of trauma, as well as its potential to influence an individual’s behavior and experiences. This knowledge can aid in the development of strategies to support individuals in minimizing their chances of acquiring HIV. Additionally, service providers need to be sensitive to the culture and to acknowledge the peer-mentor and gender-specific needs of Black/African American women living with HIV. 

Although findings of the vocational training target social determinants of health in Black/African American women living with HIV, in this case, the impact of poverty needs more exploration. This study provides supportive evidence of the positive impact of training on health, psychosocial, financial/legal, and vocational development associated with work decision making. Therefore, service providers are encouraged to investigate these four working domains when working with Black/African American women living with HIV, tailoring different considerations to support their clients through trauma recovery and vocational development. By doing so, service providers can better align their services with the holistic needs of their clientele, leading to improved health and vocational outcomes for women living with HIV.

Moreover, this study enhanced the understanding of the link between TIC principles and the considering work domains. Specifically, this study suggests that trauma-informed HIV and vocational development intervention can lead to a positive experience for Black/African American women living with HIV, further promoting health, psychosocial, financial/legal, and vocational development. Because this is the first research study that examined the impact of integrated HIV-prevention and vocational intervention using both the trauma-informed and CFCW models, more research efforts are needed with different approaches. Further application of the CFCW model for Black/African American women with other chronic health issues and/or disabilities may be considered.

Researchers and program staff need to assess the needs of their targeted population before implementing TIC into their research design, interventions, or programs and need to consider cultural and societal factors that might affect their experiences. Understanding the effects and manifestations of trauma in their specific population can help program staff tailor their approach to be more effective and sensitive to the needs of the people they serve.

Moreover, logistical factors such as resource availability, staff training, and potential partnerships with other organizations should be considered. It is not sufficient to merely add TIC components to a program; staff must be trained to recognize signs of trauma and implement trauma-informed practices, which could require additional funding or collaboration with other agencies or organizations. Lastly, the decision to integrate TIC should also be informed by the ongoing program evaluation and research outcomes to ensure its effectiveness and appropriateness for the participants being served. Integrating TIC into programs or interventions requires thoughtful consideration of the needs of the participants, adequate resources, staff training, and constant monitoring and evaluation to ensure its positive impact on the individuals served.

## 5. Limitations

The limitations of the current study were likely to be influenced by changes in program structure and different levels of participation. First, women participated in different modifications of the program structure because of the funding available at the given time. For example, some participants stayed overnight at a hotel and participated in a four-day training, whereas other training sessions took place in community centers where the length of the training was shorter. Another example of program structure changes is the group’s racial/ethnic composition. Specifically, some participants were in a single race/ethnicity group (i.e., all Black/African American), whereas others were in mixed-race/ethnicity groups, such as individuals with Hispanic or Latino backgrounds. As a result, some participants may be able to provide insights into the impact of a particular program structure that cannot be learned from other participants. 

Secondly, because participation in vocational training is not mandatory, not all participants of the current study were involved in the second phase. Women had a different level of participation in that some women were involved in all three training phases while some only completed phase one, which resulted in a lack of a more in-depth understanding of the impact of varied levels of participation in Common Threads. Another potential limitation of the study is that all research participants were volunteers. Participants who volunteered for the study may have had more positive engagements with Common Threads than other women who completed Common Threads’ training. 

## 6. Conclusions

In response to the health disparities facing Black/African American women living with HIV, some interventions were developed to address the factors that affect the disparities. The purpose of this study is to explore the impact of Common Threads, a trauma-informed integrated HIV-prevention and vocational intervention for Black/African American women living with HIV using a grounded theory analysis. The trauma-informed framework and considering work model provided background understanding to support the development of this study. Instead of quantifying the degree of impact, the goal of this study was to explore the fundamental mechanisms that contributed to the success of the program. The findings suggest that program structure and group dynamics influenced these perceived outcomes. A common refrain among participants was that the training offered an environment that facilitated a sense of safety, trust, and empowerment and that peers were honest and open rather than judgmental. Themes were also identified in participants’ perceived program outcomes. Because of the positive group features, participants could build a supportive community, gain knowledge, obtain information, manage their health, improve their perception of mental health and psychosocial skills, gain vocational skills, and consider different vocational opportunities. The findings of this study also highlight the importance of considering cultural, gender-specific, and peer-mentor needs to support Black/African American women living with HIV. This study offers several implications for practice and research. This study is essential in light of the current political climate and limited funding for HIV interventions. By exploring the influence of Common Threads on the health, psychosocial, financial/legal, and vocational development of participants, this study provides evidence supporting the effectiveness of culturally and gender-specific, trauma-informed HIV and vocational intervention components for Black/African American women living with HIV.

## Figures and Tables

**Table 1 ijerph-20-06649-t001:** Characteristics of Participants.

Characteristics	*n* (%)
Age	
25–34	1 (5)
45–54	8 (38)
55–64	11 (52)
Not Applicable	1 (5)
Age of Diagnosis	
15–24	2 (10)
25–34	10 (48)
35–44	6 (29)
45–54	3 (14)
Education	
Less Than High School	2 (10)
Some High school	3 (14)
High school graduate/GED	3 (14)
Some College	6 (29)
Associate degree	4 (19)
Bachelor’s degree	3 (14)
Household Income	
Less than USD24,999	12 (57)
USD25,000 to USD49,999	7 (33)
USD50,000 to USD99,999	2 (10)
Currently Working	
Yes	11 (52)
No	10 (48)
Working at the time of HIV diagnosis	
Yes	12 (57)
No	7 (33)
Not Applicable	2 (10)
Public Disclosure	
Yes	18 (86)
No	3 (14)
Exposure	
Heterosexual contact	16 (76)
Heterosexual contact/IDU	1 (5)
Other	1 (5)
Unsure	3 (14)
CD4 Count	
Less than 500	2 (10)
500–1200 cells/mm^3^	11 (52)
Above 1200	2 (10)
Not Applicable	6 (29)
Viral Load	
Undetectable	21 (100)
Participation Phase One Training	21 (100)
ME Circle	17 (81)
Market Place	14 (67)

GED = General Education Development. IDU = Injection Drug Use. Not Applicable = Information Missing.

**Table 2 ijerph-20-06649-t002:** Example quotes for themes related to trauma-informed program-facilitating factors.

Themes	Example Quotes
Safety	“[Women from Common Threads] didn’t sugarcoat nothing for me. But when I hurt, they held my hand. They never told me, ‘Don’t cry’, [or] ‘Stop that crying’. That’s what my boyfriends told me when I cried when they hit me. [Women from Common Threads] walked me through every difficult step of the way, every step (P21, age 58)”.
Trust and Transparency	“I have a problem with trust, but I could trust them because they were saying things, they felt like they could trust me, so I had to give it back to them. We learned to trust one another, and we all are dealing with the same thing (P6, age unknown)”.
Choice and Empowerment	“[Common Threads] educates the women and empowers them to learn how to be independent and self-sustaining. We need to have a class on teaching women how to extend their money. How to budget (P12, age 55)”.
Culture and Race	“[When I was ten years old] I didn’t think I was different until one day, a white lady walked up to me and said, “n-word, what are you doing here?” That was the first time I saw white women, and I realized that I was different. My frame of reference was that I’m in the wrong place; I should never be in a place where white folks are (P21, age 58)”.

**Table 3 ijerph-20-06649-t003:** Themes of perceived program outcomes on the considering work domains.

Considering Work Domains	Theme	Definition
Medical	Manage health	Increased awareness of personal health status, participants’ ability to care for themselves, and their consideration of medication adherence.
	Improved knowledge and resources	Attaining the knowledge and resources necessary to promote health.
Psychosocial	Building a community	Creating an environment with a shared understanding of the issues participants face and a common goal to support women living with HIV.
	Improving mental health	Discussions on how mental health had improved after participating in the program.
	Improving psychosocial skills	Discussions on different areas of improvement in psychosocial skills, including managing stress and enhancing communication skills.
Financial/Legal	Increasing financial/legal resources	A perceived change in personal development that promotes financial and legal well-being because of Common Threads.
Vocational	Vocational skills	Discussions on how participants’ vocational skills and options had expanded or strengthened.
	Considering vocational activities	Discussions on how participants began considering work or education because of Common Threads.

**Table 4 ijerph-20-06649-t004:** Example quotes related to themes of perceived program outcomes.

Themes	Example Quotes
Medical Domain: Manage Health	“We all go back and forth with this. Most people would say, ‘Oh, I don’t want to take this medicine I don’t need’, but there are certainly necessary times when people do need it. [Common Threads gave me] the motivation that if I keep building my body and keeping it healthy, I can continue to do these good things with other people who are producing (P3, age 57)”.
Medical Domain: Improved knowledge and resources	“When I meet young women with children, I would introduce them to protection, like a condom, or get them to have a pap smear every three months, [and] let them know] there are certain things you can do to check up on yourself (P6, age unknown)”.
Psychosocial Domain: Building a community	“[Common Threads] has allowed me to realize that we may look different, but we have more in common than not. We may have reached this place through different roads, arenas, behaviors, or incidents, but we’re here now, and the common thread is that you don’t have to go alone (P9, age 51)”.
Psychosocial Domain: Improving mental health	“[We have] seen these women survived and thrived. And we can do that also (P7, age 60)”. “By them doing what they did, I was able to mimic [what they did] (P4, age 51)”.
Psychosocial Domain: Improving psychosocial skills	“When I grew up, girls never said no. Now when I think I am being abused, I learned the power of saying, ‘No, I am not doing it. It doesn’t feel good, no’. I learned that saying ‘no’ is okay (P15, age 60)”.
Financial/Legal Domain: Increasing financial/legal resources	“[Common Threads] brought about ways to produce income because a lot of us may not be able to work in the secular world, but we still need to support ourselves. I was so impressed [to learn how] that a person could be financially stable. Take the skills that you have and transfer them and become your own boss (P9, age 51)”.
Vocational Domain: Vocational Skills	“[M.E. Circle] helped us to prepare for seeking employment services and taught us that we could be empowered by other women and be self-sufficient. Common Threads teaches you to love yourself, and then you can start manifesting the other stuff in your life. You can come out of fear, and you can come out of shame and guilt (P16, age 60)”.
Vocational Domain: Considering vocational activities	“I wanna go back to work. Now, I want to work for myself. I want to be an entrepreneur and teach what I do. [I need] funding and know how to write a grant. I have written a business plan but need a mentor to show me how to go about it. Someone to guide me to the right path that has done something like that before (P18, age 55)”.

## Data Availability

Data is unavailable due to privacy or ethical restrictions.

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
