# Peer review of "Exploring the Impact of an Integrated Trauma-Informed HIV and Vocational Intervention for Black/African American Women Living with HIV"

_ijerph, 2023, doi:10.3390/ijerph20176649_

Round 1

Reviewer 1 Report

Very well written manuscript that provides sufficient background knowledge, succinctly describes the methods that could be reproduced, and details the results along with explaining them in the discussion. 

4.3. Implications for Practice and Future Research: It would be advantageous to add a paragraph to describe what factors  researchers and program staff ought to consider to decide whether to integrate TIC into their interventions / programs. 

Author Response

Dear Reviewer 1,

Please see the attachment for our point-to-point responses to your comments on our manuscript “Exploring the Impact of an Integrated Trauma-Informed HIV and Vocational Intervention for Black/African American Women Living with HIV (ijerph-2481342)”.

We would like to take a moment to thank you for taking the time to thoroughly review our manuscript and for providing detailed, constructive comments that have helped us to make significant improvements.

Reviewer’s points are bolded and in italic. Our answers to reviewers’ questions are in normal text. The proposed revisions are in plain blue text. Additions are bolded in blue color with deleted content marked by strikethrough.

Thank you very much for your kind consideration of our resubmitted manuscript.

Sincerely and on behalf of all co-authors,

Hsiao-Ying (Vicki) Chang, Ph.D., CRC
Cornell University, Yang-Tan Institute on Employment and Disability

Co-authors:
Ms. Vanessa Johnson, Ribbon; [email protected]
Dr. Liza Conyers, Pennsylvania State University; [email protected]

Reviewer 2 Report

Excellent article! I highly recommend the following:

The main question addressed is the perceived effectiveness of an Trauma-informed Care intervention (Common Threads) among Black/African American women living with HIV. This is a new intervention - studies usually address what is needed in an intervention, instead of program implementation - which makes this a novel study and addressing a gap in the field. Compared with other published material subject area provides important qualitative research for a pilot to help Black/African American women living with HIV through increasing knowledge, addressing feelings of isolation.

Regarding the methodology, there should be participant information included with each quotation - further, there should be a table of quotations for each theme that includes said participant information (e.g., age)

The conclusions properly address evidence and arguments presented in the study.

 Some references could be updated - however overall appear sufficient. Would aim for references within the last 10 years as there has been more research conducted with Black/African American folks living with HIV, some of which not cited (e.g., Dr. Berkley-Patton's research)

Author Response

Dear Reviewer 2,

Please see the attachment for our point-to-point responses to your comments on our manuscript “Exploring the Impact of an Integrated Trauma-Informed HIV and Vocational Intervention for Black/African American Women Living with HIV (ijerph-2481342)”.

We would like to take a moment to thank you for taking the time to thoroughly review our manuscript and for providing detailed, constructive comments that have helped us to make significant improvements.

Reviewer’s points are bolded and in italic. Our answers to reviewers’ questions are in normal text. The proposed revisions are in plain blue text. Additions are bolded in blue color with deleted content marked by strikethrough.

Thank you very much for your kind consideration of our resubmitted manuscript.

Sincerely and on behalf of all co-authors,

Hsiao-Ying (Vicki) Chang, Ph.D., CRC
Cornell University, Yang-Tan Institute on Employment and Disability

Co-authors:
Ms. Vanessa Johnson, Ribbon; [email protected]
Dr. Liza Conyers, Pennsylvania State University; [email protected]

Reviewer 3 Report

What the authors have here is a great phenomenological study presenting data on the impact of the Common Threads intervention on women living with HIV.  What is described here is not a grounded theory approach, nor do the authors utilize grounded theory methods in coding and analyzing the data and utilizing data to generate theory.  Given this, the paper is not publishable as presented.  I encourage the authors to reconsider the presentation of these data as what you have outlined provides great support for this intervention.  Throughout the manuscript there are incomplete sentences, long and confusing sentences and spelling and grammatical errors.  Some other things to consider:

Introduction:

Page 2, lines 47-57 - what the authors are describing here are syndemics.  Consider incorporating that work here to inform your work - see Millett and Poteat's publications in this area.

Page 2, lines 88-90 - insurance coverage is tied to employment in this country so your argument re: full time work ignores this important fact.

Methods:

Page 4, lines 200-203 - you need a citation here for any publications you have describing Common Threads.  If you have none, consider turning this into a pub that describes the intervention and the long-term impact on women living with HIV.

Page 5, lines 208 - what do you mean by "disclosed publicly"?  Is this HIV status disclosure?  If so, what is public as there are many different types of "public disclosure" among friends, family, etc.

The authors assume that there is a common understanding of Trauma Informed Care.  Need to describe it and provide citations.

What you are describing here is more of a phenomenological approach to data collection and analysis. With grounded theory you need to explain how you analyzed each interview to build on your hypothesis and then made modifications to obtain more information to get you closer to your theory.  A detailed description of how the interview guide was developed and then refined as data was acquired is needed in a GT approach.  Coding of transcripts happens as you complete them, not in batches.

Results:

Page 10, lines 376-377 - you state that women had already begun practicing healthier lifestyles BEFORE joining Common Threads.  This is important.  Why?  Are there other factors to be considered for this population?

The Financial/Legal domain data do not appear to be distinct from the vocational domain data.

Discussion:

This section is where you go into depth about your results.  It is not where you bring in new data not discussed in the results (as on page 14, lines 615-617) or where you repeat data verbatim (as on page 14, lines 627-629).  There are entire chunks that are repetitive (as on page 18, lines 800-828).

Limitations:

In qualitative data participant memory and experiences are the data - not a limitation of the data.  The things you list as influencing memory and experience are exactly what you are taking into consideration when you are sampling to ensure that you are obtaining diverse perspectives.

Purposive sampling is also not a limitation.  It is a strategy utilized in qualitative data collection to ensure that the people in your sample have experienced a specific phenomenon.

Did you see any differences in the impact of Common Threads on women with different levels of participation?  This would be interesting and informative. 

There are spelling and grammatical errors throughout.  I would suggest a thorough review for language.

Author Response

Dear Reviewer 3,

Please see the attachment for our point-to-point responses to your comments on our manuscript “Exploring the Impact of an Integrated Trauma-Informed HIV and Vocational Intervention for Black/African American Women Living with HIV (ijerph-2481342)”.

We would like to take a moment to thank you for taking the time to thoroughly review our manuscript and for providing detailed, constructive comments that have helped us to make significant improvements.

Reviewer’s points are bolded and in italic. Our answers to reviewers’ questions are in normal text. The proposed revisions are in plain blue text. Additions are bolded in blue color with deleted content marked by strikethrough.

Thank you very much for your kind consideration of our revised manuscript.

Sincerely and on behalf of all co-authors,

Hsiao-Ying (Vicki) Chang, Ph.D., CRC
Cornell University, Yang-Tan Institute on Employment and Disability

Co-authors:
Ms. Vanessa Johnson, Ribbon; [email protected]
Dr. Liza Conyers, Pennsylvania State University; [email protected]

Round 2

Reviewer 3 Report

The authors were incredibly responsive to the feedback provided.  The manuscript is much improved.  All of the inaccuracies have been corrected.  Great work!  This is important work, and I am pleased that you were able to make significant improvements.